# Transcriptome Changes in the *Mink* Uterus during Blastocyst Dormancy and Reactivation

**DOI:** 10.3390/ijms20092099

**Published:** 2019-04-28

**Authors:** Xinyan Cao, Jiaping Zhao, Yong Liu, Hengxing Ba, Haijun Wei, Yufei Zhang, Guiwu Wang, Bruce D. Murphy, Xiumei Xing

**Affiliations:** 1Institute of Special Animal and Plant Sciences, Chinese Academy of Agricultural Sciences, #4899 Juye Street, Jingyue District, Changchun 130112, China; xinyan_99@163.com (X.C.); tcszjp@126.com (J.Z.); Bahengxing@caas.cn (H.B.); weihaijun2005@sina.com (H.W.); Zhangyufei@caas.cn (Y.Z.); wangguiwu2005@163.com (G.W.); 2Key Laboratory of Embryo Development and Reproductive Regulation of Anhui Province, College of Biological and Food Engineering, Fuyang Teachers College, Fuyang 236000, China; drliuyong@gmail.com; 3Centre de Recherché en Reproduction et Fertilité, Faculté de Médicine Vétérinaire, Université de Montréal, St-Hyacinthe, QC J2S 2M2, Canada

**Keywords:** embryo diapauses, activated, RNA-seq, uterus, *mink*

## Abstract

Embryo implantation in the *mink* follows the pattern of many carnivores, in that preimplantation embryo diapause occurs in every gestation. Details of the gene expression and regulatory networks that terminate embryo diapause remain poorly understood. Illumina RNA-Seq was used to analyze global gene expression changes in the *mink* uterus during embryo diapause and activation leading to implantation. More than 50 million high quality reads were generated, and assembled into 170,984 unigenes. A total of 1684 differential expressed genes (DEGs) in uteri with blastocysts in diapause were compared to the activated embryo group (*p* < 0.05). Among these transcripts, 1527 were annotated as known genes, including 963 up-regulated and 564 down-regulated genes. The gene ontology terms for the observed DEGs, included cellular communication, phosphatase activity, extracellular matrix and G-protein couple receptor activity. The KEGG pathways, including PI3K-Akt signaling pathway, focal adhesion and extracellular matrix (ECM)-receptor interactions were the most enriched. A protein-protein interaction (PPI) network was constructed, and hub nodes such as VEGFA, EGF, AKT, IGF1, PIK3C and CCND1 with high degrees of connectivity represent gene clusters expected to play an important role in embryo activation. These results provide novel information for understanding the molecular mechanisms of maternal regulation of embryo activation in *mink*.

## 1. Introduction

Embryo implantation is a dynamic process, that comprises periods of uterine pre-receptivity, receptivity and implantation phases, involving morphological and molecular changes in the endometrium. This process varies among mammals. *Mink* are within the subset of carnivores in which pre-implantation embryo diapause occurs in every gestation [1,2]. It is believed that the period of embryo arrest in the uterus is associated elevated embryonic mortality rates. Reciprocal embryo transfer between *mink* and the ferret, a related species exhibiting no embryo diapause, revealed that diapause is controlled primarily by the maternal uterine environment, rather than by the embryos [3]. When diapause embryos are co-cultured with endometrial epithelial and stromal cells, they show extended survival of the embryos in vitro and, in some cases, escape from their quiescent state [4]. The molecular changes in the uterus during implantation have been explored [5,6,7]. However, the precise factors and mechanisms by which the uterus activates the embryo from diapause remain to be elucidated.

Previous studies of differential gene expression in the uterus during reactivation of embryo development in *mink* were carried out using the suppressive subtractive hybridization (SSH) technique, generating a library of 123 differential expressed genes (DEGs) between uteri with blastocysts in diapause and uteri with reactivated blastocysts [7]. This method has a number of limitations, and does not to provide a comprehensive analysis of the global transcriptome. With the development of high-throughput mRNA sequencing technology (RNA-seq), Illumina platform has become the most used method, as it is an accurate tool for quantifying RNA abundance and diversity, and has the capacity to identify novel transcribed regions, splice isoforms and single nucleotide polymorphisms (SNPs). Indeed, a number of studies have employed the platform to understand the dynamics of gene expression during early embryonic development. For example, Zhang [8] using the RNA-Seq method, found that 3255 unigenes were differentially expressed between the receptive and pre-receptive endometria in *dairy goats*. In the *porcine* endometrium, Samborski [9] reported 2593 DEGs between pregnant and non-pregnant animals. Furthermore, differential gene function, multiple pathways and regulatory networks were also identified in these studies. 

Prenatal mortality is a constraint on litter size in *mink*. It has been well demonstrated that embryonic losses occur during gestation, and it is believed that much of this mortality occurs during diapause in *mink*. It is estimated that approximately 60% of fertilized ova do not survive through gestation in this species [1,10]. A detailed investigation into global gene expression in the maternal uterine environment during the establishment of pregnancy will augment understanding of gene function and of the regulatory mechanisms that influence embryonic survival and implantation. This information will be valuable for the development of mitigation strategies to increase the frequency of successful pregnancy establishment and larger litters at parturition. 

Here, we present a wide screen RNA analysis to characterize gene expression of uterine tissue during embryo diapause and reactivation. We adopted the Illumina sequencing approach to obtain a larger and more reliable transcriptomic dataset, to provide information on candidate regulatory factors, and illustrate intricate molecular regulatory networks and biological functions that characterize the uterine changes during the transition from diapause to embryo activation in the *mink* uterus. 

## 2. Results

### 2.1. Identification of Transcriptomic Differences

This study used RNA-Seq to investigate the transcriptome changes of the uterus from diapause to the activation period of *mink*. RNA was purified from the uterus in diapause and activation phases with three biological replicates for each group. Sequencing of the libraries yielded more than 54.3 and 57.6 million quality reads for the diapause and activated group, respectively. After removing invalid reads, we acquired 52.1 and 55.3 million clean reads from each group, respectively. A total of 170,984 unigenes were obtained from clean reads using the optimized parameters. 

A total of 1684 transcripts were identified to as being differentially expressed (based on log2 fold change, *p* < 0.05) in whole uterine horns between the diapause and activated groups. Among these transcripts, 1527 were annotated as known genes, thus approximately 10% of transcripts were unable to be annotated. Among the DEGs, 963 were significantly more abundant in the activated uterus (622 genes fold > 2, and 167 genes fold > 1.5 and simultaneously fold < 2), while 564 genes were more abundant in diapause groups (445 genes fold > 2, and 58 genes fold > 1.5 and simultaneously fold < 2) (Table 1). The top 20 most DEGs (highest *p* value) in the activated uterus are presented in Table 2. 

### 2.2. Gene Ontology Analysis of the DEGs

To gain insight into the biological functions that are regulated during embryo activation, we conducted a function enrichment analysis of DEGs based on gene ontology (GO). A total of 241 GO terms were overrepresented between the activated and diapause uteri (*p* < 0.05). Of the 148 GO terms that were identified in the biological process, the most significantly enriched terms were the xenobiotic metabolic process (20 genes), cellular response to stimulus (219 genes), cell communication (190 genes), signal transduction (186 genes), G-protein coupled receptor signaling pathway (42 genes), regulation of embryonic development (7 genes), vitamin metabolic process (24 genes), regulation of cell adhesion (18 genes), regulation of cell migration (32 genes), regulation of locomotion (8 genes) and immune response (20 genes). In the cellular compartment GO category, 17 terms were significantly enriched. The most significantly enriched GO terms were extracellular matrix (9 genes), extracellular region part (45 genes), collagen trimer (9 genes), membrane (318 genes), interleukin-1 receptor complex (1 genes) and MHC protein complex (5 genes). In the molecular function, phosphatase activity (38 genes), N,N-dimethylaniline monooxygenase activity (5 genes), G-protein coupled receptor activity (27 genes) and metallocarboxypeptidase activity (5 genes) were the most enrichment (Table 3). The most significantly enriched pathways were responsible for endometrial morphological changes and function involved in the extracellular matrix, cell migration and adhesion. Genes included laminin (*LAMA1, 3, 5*), collagen (*COL1A1, COL4A, COL12A1, COL6A*), *ADAM* metallopeptidase (*ADAMTS9, ADAMTS12, ADAM33*), tropomyosin (*TPM1*), osteonectin (*SPARC, SPOCK*) and thrombospondin (*THBS1, THBS2*).

### 2.3. KEGG Pathway Analysis of DEGs

KEGG pathway enrichment analysis was performed on these DEGs and a cut-off criterion of *p* < 0.05 was also used. A total of 18 enriched pathways were found between the activated and diapause groups. The classification indicated that the *PI3K-Akt* signaling pathway (Figure 1), focal adhesion, ECM-receptor interaction (Figure 2), cell adhesion molecules and protein digestion and absorption were highly enriched. The top 10 KEGG pathways are shown in Table 4. The most significantly enriched pathways were those responsible for cell adhesion and those involved in extracellular matrix remodeling, highly enriched genes included *CD44*, *LAMA1–5*, *ITGB4*, *ITGA5*, *ITGB8*, *COL4A*, *ITGAM* and *LAMC3*. Genes in the pathways related to immune gene expression, such as *COL1AS, COL6A, FGFR4, XPNPEP2, IL1A* and *MHC2* were likewise enriched. 

The pituitary hormone, prolactin, has recently been shown to have direct effects on the *mink* uterus, acting through membrane receptors [2], promoting epithelial cell changes via janus kinase-2 signal transducer and activated *PI3K-Akt* signaling pathway (Figure 3). This pathway has been implicated in the regulation of cell survival, apoptosis, cell cycle progression, metabolism, and cell proliferation and growth. Genes enriched in this pathway include *ELF5, PIK3C, AKT, CYP17A, PRLR, STAT5A* and *CCND1* (Table 4). 

### 2.4. PPI Network Analysis

Protein-protein interactions (PPIs) analysis can reveal the protein function of DEGs at the molecular level. In the present study, we used the STRING database to construct a gene network (Results in Appendix A). This network consisted of 354 nodes connected via 697 edges. The highly connected nodes, also known as hub genes, represent important genes in the network. We identified 6 hub nodes with a high connectivity degree >20 (Figure 4). The genes were *AKT* (41), *EGF* (36), *P1K3C* (25), *IGF1* (24), *CCND1* (22) and *VEGFA* (21). The data suggest that they play important roles in embryo activation in *mink*. 

### 2.5. Candidate Gene Selection and Validation of Expression Levels

We used qPCR to validate 21 DEGs from the RNA-Seq analysis that have also been reported to be relevant to the physiology of implantation. Validation experiments were performed on RNA samples from biological replicates independent of the animals used for RNAseq analysis. As shown in Figure 5, the expression levels of *EGF, HBEGF, ER, IGF1, LIF, FKBP4, ODC, ASMT, PDE11, MCM2, PIK3C2, SLC4A8, CD1, VEGFA, c-myc, PRLR* and *LAMA3* were elevated in the activated relative to the diapause uterus. The largest differences were 22-, 17- and 14-fold increases of the *LIF, FKBP4, VEGFA* transcripts, respectively. In contrast, the abundance of the *EGFR, PMP22, GALNT* and the progesterone receptor was reduced in the activated uterus. Two candidate genes, *LIF* and *ESR1,* did not show a significant difference in RNA-Seq result but differences were detected by qPCR. 

### 2.6. Prolactin Activates Diapauses Embryo by PI3K Signaling Pathway

To explore the role of *PI3K/AKT* pathway in the escape of embryo diapauses in *mink*, we cultured *mink* embryos with or without pathway inhibitor LY294002 (10 µM, 100 µM), Adding inhibitor into the embryo culture medium over five days significantly decreased the percentage of embryo survival and the diameter of surviving blastocysts, compared to those in the control group without inhibitor supplementation (*p* < 0.05, Figure 6). 

## 3. Discussion

The *mink* is one of a large number of carnivores in which arrested development at the blastocyst stage occurs for a varying period of time [11]. In this species, the minimum duration of diapause is approximately six days and the average duration about 20 days. It is believed that the embryo is vulnerable during this period, and embryo loss during diapause contributes to gestational failure. Embryo transfer experiments have confirmed that embryo diapause is controlled by the uterine environment, and that developmental arrest is due to a lack of specific factors necessary for continued embryo development [3]. Establishing the crosstalk between the blastocyst in diapause and the uterus involves a number of complex signalling networks. In this study, we present the first application of RNA-Seq to characterize the uterine transcriptome during embryo diapause and activation in *mink*. More than 50 million clean reads were identified, and 170,984 unigenes were obtained from the clean reads using optimized parameters. A total of 1684 DEGs were generated for activated compared to uteri in diapause. Among these DEGs, we selected 21 to validate the accuracy of RNA-Seq by qRT-PCR, and found that the expression trend of these genes measured by qRT-PCR was consistent with RNA-Seq, suggesting that our data were of high quality. 

Termination of embryonic diapause in this species is induced by prolactin from the pituitary gland, followed by ovarian luteal cell activation and secretion of progesterone, subsequently acting on the endometrium [12]. Prolactin receptor (*PRLR*) has been found in the uterus of *mink*. In our RNA-Seq results, the *PRLR* was significantly increased (about 16-fold) in the activated compared to the diapause uterus. A recent report demonstrated that prolactin induces expression of factors essential for the termination of diapause in the *mink* [2]. *PRLR* deficiency resulting in implantation failure has been reported in *mice* [13]. The study showed that ovarian *PRLR* is required at implantation, while uterine *PRLR* may be essential for supporting late gestation. Previous studies have reported that prolactin, acting through its cognate receptor, exerts its role via the Jak2-Stat signaling pathway [14,15]. Other pathways, for example, tyrosine phosphorylation of phosphatidylinositol (PI)-3′ kinase (*P13K*) are also involved in signal transduction by this receptor [16]. Our global transcriptome analysis indicates both of these intracellular signalling cascades are upregulated in the activated uterus, compared to the diapause counterpart. The existence of two *PRL*-dependent signalling cascades is initiated by the c-Src-mediated activation of *Fak/Erk 1/2* and *P13K* pathways that control the expression of c-Myc and cyclin D1 and, consequently, cell proliferation [17]. The *P13K-Akt* pathway is a well-known mediator of cell growth, proliferation, migration and angiogenesis [18,19,20,21]. During embryo implantation, the *P13K-Akt* pathway is involved in trophoblast invasion, immune surveillance and apoptosis [22,23,24]. In the present study, the signalling pathway was the most enriched KEGG pathway, in that 24 DEGs were upregulated and 11 DEGs were down-regulated in the activated uterus compared to the diapause uterus. Adding an inhibitor of this pathway during in vitro culture of embryos inhibited their development. Further investigation of downstream targets of these pathways is required to determine the uterine genes that regulate the reactivation of the *mink* embryo from diapause. 

The up-regulated DEGs with the highest expression levels associated with activation were Ankyrin repeat and SAM domain-containing protein (*ANKSIA*). Ankyrin repeat proteins are found with diverse functions, which can be attributed to the presence of catalytic domains in various ankyrin repeat-containing proteins, for example, SH3, SAM and PDZ domains in addition to seven ankyrin repeats. These proteins are among the most frequently observed amino acid motifs in protein databases, and have received a great deal of attention recently. Defects in ankyrin repeat proteins has been found in a number of *human* diseases [25]. Our qRT-PCR validation indicated that the abundance of this transcript was significantly higher in the activated uterus compared to the diapause phase, however, little is known concerning the relationship between *ANKSIA* and embryo activation in *mink*. Therefore, we propose that *ANKSIA* may be positively involved in development of the receptive endometrium from the static state in *mink*. 

Laminin alpha 3 (*LAMA3*) is a member of a secreted family of heterotrimeric molecules essential for basement membrane formation, cell migration and mechano-signal transduction. *LAMA3* has been identified in the uterus of several species during implantation, and decreased expression may be associated with reduced uterine receptivity and infertility [26,27,28]. In the *pig*, RNA-Seq analysis in endometrial tissues during early pregnancy showed that *LAMA3* expression is upregulated during embryo implantation [26] and the present study demonstrated, for the first time, higher expression of *LAMA3* in the activated uterus of *mink*. These findings suggest that uterine *LAMA3* expression is regulated in a stage-specific manner during early pregnancy, and that *LAMA3* may have a critical role in regulating embryo activation.

The DNA replication licensing factor *MCM2* (Minichromosome maintenance protein 2), which is essential for eukaryotic DNA replication, is dynamically expressed in both proliferative and differentiated stromal cells in the *mouse* peri-implantation uterus and experimental reduction of *MCM2* compromises stromal cell differentiation [29]. Studies using both in vivo and in vitro approaches demonstrated that the *MCM2* protein is regulated by estrogen induced *KLF4* (Kruppel-like family) expression, as progesterone levels increase, *KLF15* is induced, and the abundance of *MCM2* declines in uterine stromal cells [30]. Ovarian progesterone secretion increases at reactivation and levels of both progesterone and estradiol being high around implantation [31,32]. However, expression of neither progesterone nor estrogen receptors displayed increases in reactivated embryo and uterus in *mink* [2]. In the *mink*, prolactin is the main regulator of embryo implantation and the prolactin receptor expressed increases around the time of implantation [33,34,35]. Progesterone and/or estrogen does not induce implantation in *mink*, although they are sufficient to induce reactivation from diapause in the mouse and a number of other species [2]. This study showed that the progesterone receptor expression declined, while *MCM2* expression increased during embryo activation in *mink* uterus, suggested a different mechanism with other mammals. 

Extracellular matrix (*ECM*)-receptor interaction and focal adhesion were the most predominant altered pathways in activated compared with diapause uteri. Previous studies have reported that a variety of *ECM* proteins were present on the surface of uterine stromal cells [28,29], and these cell surface glycoproteins function in epithelial-embryonic interactions. The levels of a number of ECM molecules have been shown to peak during the “window of implantation” in *mice* [36]. In *mink*, we found 19 DEGs enriched into the pathway of *ECM*-receptor interaction, and 28 DEGs enriched into the focal adhesion pathway (*p* < 0.05). Although there has been extensive study of *ECM* in the uterus, the pathway of focal adhesion and interaction with *ECM* during embryo activation in the *mink* uterus is not well understood. 

We also investigated the immune response between the diapause and activated uterus based on the importance of immunity in early maternal, fetal interactions. Several genes were identified, which may be involved in this process. Interleukin-1 (*IL1A*), which triggers cell activation via the functional signaling *IL1R1. IL1* was shown to increase tumor invasiveness and metastasis by enhancing the expression of adhesion molecules on endothelial [37]. *IL1* also stimulates the proliferation of endothelial cells and production of cytokines [38,39]. Besides its major immunological effects, *IL1* regulates cell growth and angiogenesis by activating the *VEGF-KDR* pathway [38,40]. Early studies indicated an essential role of *IL1* in implantation, as repeated injections of *IL1* receptor antagonist into pregnant mice prior to implantation caused implantation failure [41]. In *cattle*, *IL1A* stimulates *PGF2α* synthesis and secretion by endometrium in vitro and in vivo [42,43]. *PGs* are necessary for increased vascular permeability at the site of implantation [44] and are implicated in decidualization of rodent and human stromal cells [45,46]. In *mink*, expression of PTGS2, which is the regulated enzyme in *PG* synthesis, has been reported to be a transient event that occurs at the time of trophoblast attachment and invasion [5]. In addition, the genes with greater abundance in the activated compared with diapauses uterus, such as *PRIMA1* (proline rich membrane anchor 1), *GDF7* (growth differentiation factor 7), *AGPAT9* (glycerol-3-phosphate acyltransferase 3), *ATRNL1* (attractin-like 1), *TNFSF18* (necrosis factor), *COL1A1* (collagen type I), *PHLPP1* (PH domain and leucine rich repeat protein phosphatase 1) and *SLC9A8* (cation proton antiporter 8) are associated with the immune response. Further studies are required to elucidate their potential roles. 

In the innate immune pathways such as cytokine-cytokine receptor interaction, cytokines modulate the endometrial receptivity by regulating the expression of various adhesion molecules [47]. In mammals, deregulated expression of cytokines and their signaling leads to an absolute or partial failure of implantation and abnormal placental formation [48]. The *CSF* family (colony stimulating factor) of ligands and their receptors are increasingly recognized to play important roles in promoting proliferation of trophoblast cells, and have recently been used as therapeutic targets for recurrent implantation failure and pregnancy loss. Activin-like kinase 6 (*ALK6*, *BMPR1B*), is known to mediate BMP signalling. *BMPR1B* has been demonstrated to play an essential role in regulation of endometrial function and female fertility in *mice* and *humans* [49]. Tumor necrosis factor ligand superfamily member 18 (*TNFSF18*) are major proinflammatory cytokines that are known to induce structural and functional changes in endothelial cells. Adhesion between trophoblasts and endothelial cells can be increased by *TNF* [50]. All these cytokines are overexpressed in the activated relative to the diapause uterus. The genes with greater abundance in the activated uterus, such as *VEGFA, PRLR* and *EGF*, were also clustered into the pathways of cytokine-cytokine receptor interactions.

Using the STRING database, we obtained a network consisted of 354 nodes and 697 edges. In this network, we identified 6 hub genes. The hub genes were *VEGFA* (vascular endothelial growth factor A), *EGF* (epidermal growth factor), *AKT* (serine/threonine-protein kinase), *PIK3C* (phosphatidylinositol-4,5-bisphosphate 3-kinase, PI3K), *IGF1* (insulin-like growth factor 1) and *CCND1* (cyclin D1). VEGF is best known for its potent endothelial cell specific mitogenic activity, and plays a role in increasing vascular permeability [51]. Investigation of *VEGF* has shown that it is expressed during the peri-implantation period in *mink* [52], and the abundance of mRNA of *VEGF* isoforms was highest during embryo activation and at implantation, which is consistent with our RNA-Seq and Q-PCR result. *EGF* has been demonstrated to be a molecular marker in endometrial receptivity [53]. Its function may be to protect endometrial integrity, as *EGF* was decreased in human endometrium in women with luteal phase defects and pregnancy loss [54,55]. *IGF1*, which promoted the proliferation of endometrial cells, was expressed in all components of the *mouse* uterus, and its synthesis was regulated by estrogen acting through its nuclear receptor [56]. *IGF1* knockout mice lack a uterine proliferative response to estrogen, specifically, lack G2/M progression of the epithelial cells [57]. It is worth mentioning that the hub genes *PI3K/AKT* were significantly upregulated in the activated uterus compared to the diapause phase consistent with the known role of prolactin in induction of implantation in this species. This is in agreement with our description in the KEGG analysis. *PI3K* proteins are key regulators involved in a wide variety of cellular processes, including endothelial proliferation, migration, differentiation and survival [58,59]. Exposure of the uterine lumen to an *AKT* inhibitor prior to embryo transfer induced early pregnancy defects, ranging from implantation failure to aberrant spacing of implantation sites [60]. *PI3K* knockout mice exhibited impaired migration of endothelial cells and subsequent loss of angiogenic activity [61]. 

## 4. Materials and Methods

### 4.1. Animals and Sample Collection

All animal procedures were conducted under guidelines of the Experimental Animal Use and Care Committee of the Institute of Special Animal and Plant Sciences, Chinese Academy of Agricultural Sciences and pre-approved prior to implementation (permit no. ISAPSWAPS2016000302. 2016/5/20). Female *mink* were mated to two fertile males, at 7–9 day intervals, according to the usual commercial farming procedures. Successful matings were confirmed by the presence of motile spermatozoa in vaginal smears.

*Mink* were subjected to general anesthesia prior to collection of samples. Uterine horns from females during diapause were collected prior to March 21 and 7–9 days after the final mating. Termination of embryo diapause was performed by injecting 1 mg kg^−1^ day^−1^ ovine prolactin (Sigma-Aldrich, L6520, Saint Louls, USA) each day beginning on March 21 and for the following five days [12]. The first day of prolactin injection was designated day 1 after blastocyst reactivation. Unimplanted blastocysts were flushed from the uterus using TC-199 medium (Gibco, 11150-067, Carsbad, USA) supplemented with 10% fetal bovine serum (Gibco, 16000-044, Carsbad, CA, USA). Blastocyst diameters were measured to confirm that embryos were activated, as described previously [12]. For transcriptome analysis, uterine horns were collected during diapause and on the sixth day after prolactin-induced reactivation, and stored in liquid nitrogen until subsequent transcriptome analysis.

### 4.2. RNA Sequencing

Total RNA extracted from uterus samples using TRIzol^®^ reagent (Invitrogen, *n* = 3 for each diapause and activated group). RNA integrity was evaluated by Agilent Bioanalyzer 2100 system (Agilent Technologies, Santa Clara, CA, USA), RNA degradation and contamination was detected on 1% agarose gels. RNA concentration was determined using the Qubit^®^ 2.0 Fluorometer (Life Technologies, South San Francisco, CA, USA), and purity was determined by NanoPhotometer^®^ spectrophotometer (IMPLEN, California, CA, USA). A total amount of 1.5 µg RNA per sample was used as input material for the RNA sample preparations. Sequencing libraries were constructed using the NEBNext^®^ Ultra^TM^ RNA Library Prep Kit for Illumina^®^ (NEB, Boston, MA, USA) following the manufacturer’s recommendations, and index codes were added to attribute sequences to each sample. The clustering of the index-coded samples was performed on a cBot Cluster Generation System using TruSeq PE Cluster Kit v3-cBot-HS (Illumia) according to the manufacturer’s instructions. After cluster generation, the library preparations were sequenced on an Illumina Hiseq platform and 150 bp paired-end reads were generated. Raw data (raw reads) of fastq format were first processed through in-house perl scripts. In this step, clean data (clean reads) were obtained by removing reads containing adapter, reads containing poly-N and low quality reads from raw data. All the downstream analyses were based on clean data of high quality. Transcriptome assembly was accomplished based on the left.fq and right.fq using Trinity [62] with min_kmer_cov set to 2 by default and all other parameters set default. Gene function was annotated based on the following databases: Nr (NCBI non-redundant protein sequences), Nt (NCBI non-redundant nucleotide sequences), Pfam (Protein family), KOG/COG (Clusters of Orthologous Groups of proteins), Swiss-Prot (A manually annotated and reviewed protein sequence database), KO (KEGG Ortholog database) and GO (Gene Ontology). Differential expression analysis of two groups was performed using the DESeq R package [63]. The Benjamini and Hochberg approaches were used to adjust *p* values and control the false discovery rate. *p* < 0.05 in DESeq were assigned as differentially expressed.

### 4.3. Enrichment Analysis of DEGs

Gene Ontology (GO) enrichment analysis of the DEGs was performed by GOseq R packages based Wallenius non-central hyper-geometric distribution [64]. KOBAS [65] software was used to test the statistical enrichment of DEGs in KEGG pathway. The threshold for the DEGs was set as *p* < 0.05.

### 4.4. Validation of DEGs

Total RNA was extracted from uteri in diapause and activation phases with Trizol (Invitrogen, 15596018, Carlsbad, USA) according to the manufacturer’s specifications. The yield of RNA was determined using a NanoDrop 2000 spectrophotometer (Thermo Scientific, Wilmington, NC, USA), and the integrity was evaluated using agarose gel electrophoresis stained with ethidium bromide. Reverse transcribed was performed using PrimeScript^TM^ RT reagent kit (TAKARA, RR036Q, Kyoto, Japan) with 10 µL reaction system, including 2 μL 5xPrimeScript Buffer, 0.5 μL PrimeScript RT Enzyme Mix l, 0.5 μL Oligo dT Primer (50 μM), 0.5 μL Random 6 mers (100 μM), 2 μL total RNA and 4.5 μL RNase Free dH_2_O and stored in −20 °C. Real-time PCR was determined using a LightCycler^®^ 480 ІІ Real-time PCR Instrument (Roche, Basilea, Switzerland) with 20 μL PCR reaction mixture that included 10 μL 2×SG Fast qPCR Master Mix (Sangon Biotech B639271, Shanghai, China), 0.4 μL of forward primer, 0.4 μL of reverse primer, 2μL DNF buffer, 1 μL of cDNA and 6.2 μL PCR-grade water. Reaction condition was 95 °C for 3 min, 95 °C for 3 s for 40 cycles and 60 °C for 30 s. Each sample was run in triplicate for analysis. In the end of the PCR cycles, melting curve analysis was performed to validate the specific generation of the expected PCR product. The expression levels of mRNAs were normalized to GAPDH and were calculated using the 2^−ΔΔ*C*t^ method [66].

### 4.5. PPI (Protein Protein Interaction) Network Construction

The sequences of DEGs were determined by BLASTX using the genome of a related species. The protein-protein interaction of which exists in the STRING database (http://string-db.org/) was employed to get the predicted PPI of these DEGs. Then, the PPI of these DEGs were visualized in Cytoscape [67]. In the PPI network, nodes stand for proteins, and edges represent interactions between 2 proteins.

### 4.6. Embryo Culture

Blastocysts in diapause were collected by uterine flushing and washed three times in TCM199 medium supplementation with 10% (*v*/*v*) fetal bovine serum (Invitrogen). Embryos were cultured in groups of 10–12 in 400 µl M16 Medium (Sigma-Aldrich, M7292,Saint Louls, MO, USA) supplementation with 10 µg/mL prolactin at 37 °C under 5% CO_2_ in humidified air, depending on the experiment, concentrations of 0, 10 or 100µM solutions of LY294002 were prepared in dimethyl sulfoxide (DMSO) and diluted in culture medium to the desired final concentration. DMSO was added to control culture, and all examined culture droplets contained 0.1% DMSO. Medium was changed every two days, and embryo diameters were measured by ocular micrometer after 5 days. Determination of death of embryos was based on observation of shrinkage of the blastocyst, blastocele size reduction or no blastocele observed. Experiments were repeated at least three times.

### 4.7. Statistical Analyses

For GO and pathway enrichment analysis, *p* < 0.05 was considered to indicate a statistically significant difference. *p*-values were adjusted using the false discovery rate (FDR) method for multiple hypothesis testing. FDR < 0.05 was established as the threshold value. For embryo culture experiment, data were analyzed by ANOVA, using statistics package for social science (SPSS) software. Percentage data were arc-sine transformed, a Duncan multiple comparison test was used to locate differences. For the qPCR analysis, a two-tail unequal variance student *t* test was used to compare the messenger RNA (mRNA) expression levels between diapause and reactivation uterus.Data were expressed as mean ± S.E.M and *p* < 0.05 was considered significant. 

## 5. Conclusions

The comparison of the transcriptome between uteri during diapause and following reactivation of the embryo presented here indicates that multiple regulatory pathways are activated during the escape of the from diapause. Hormones provide a suitable physiologic milieu for embryo implantation. Acquired diseases or alteration in maternal physiology during embryo implantation can affect endometrial function and fetal development. The prolactin signaling pathway and its downstream PI3K/AKT pathway are the most enriched pathways in activated compared to diapause uterus, blocking the pathway with inhibitor decreased embryo development, our previous study have demonstrated the importance of these pathways on embryo invasive in *mink* [68]. Further investigation is needed to amplify and validate the current study with details of the localization and function of important genes and proteins, which may allow us to better understand what is required to drive embryo development and help develop new strategies for promoting successful pregnancy in *mink*.

## Figures and Tables

**Figure 1 ijms-20-02099-f001:**
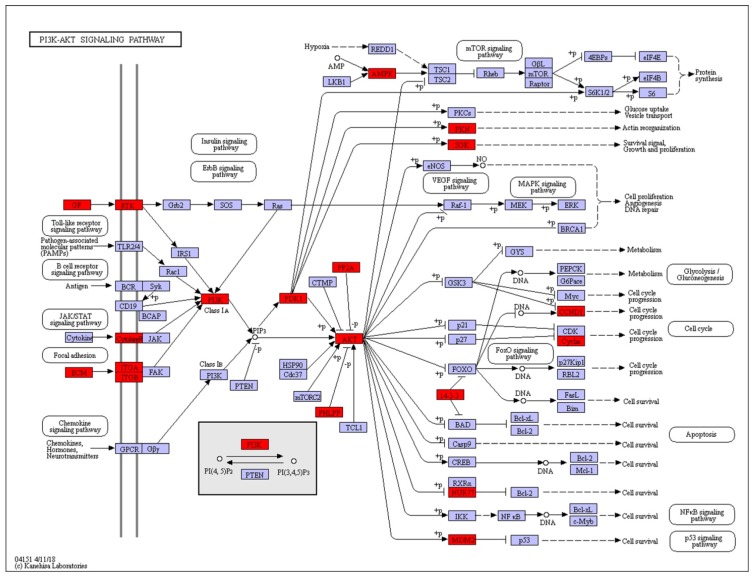
Model of activation of the *PI3K/AKT* signalling pathway in *mink* uteri in activated versus diapause states. Differentially expressed genes are shown in red color. Genes in blue boxes were present in the *mink* transcriptome but not differentially expressed.

**Figure 2 ijms-20-02099-f002:**
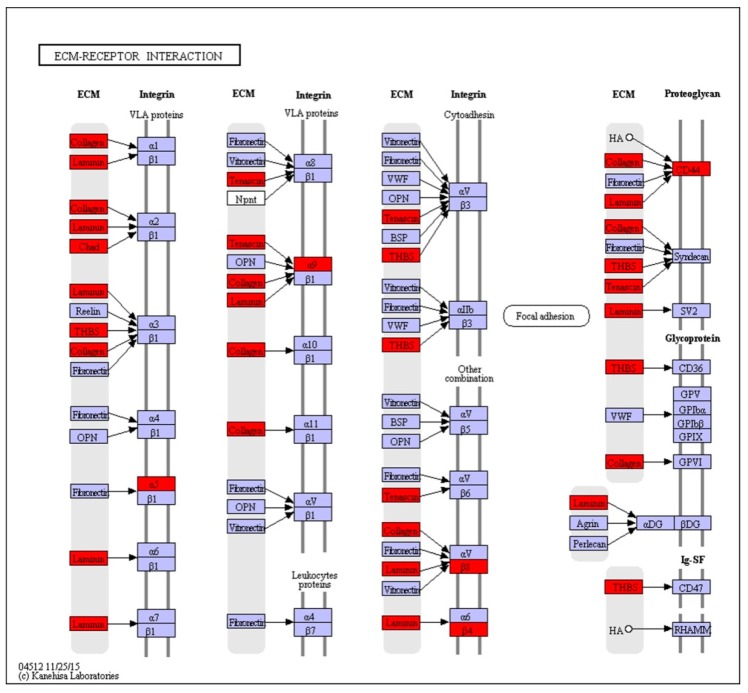
Model of activation of the extracellular matrix (ECM)-Receptor interaction pathway in the *mink* uterus in activated versus diapause states. Differentially expressed genes are shown in red color. Genes in blue boxes were present in the *mink* transcriptome but not differentially expressed.

**Figure 3 ijms-20-02099-f003:**
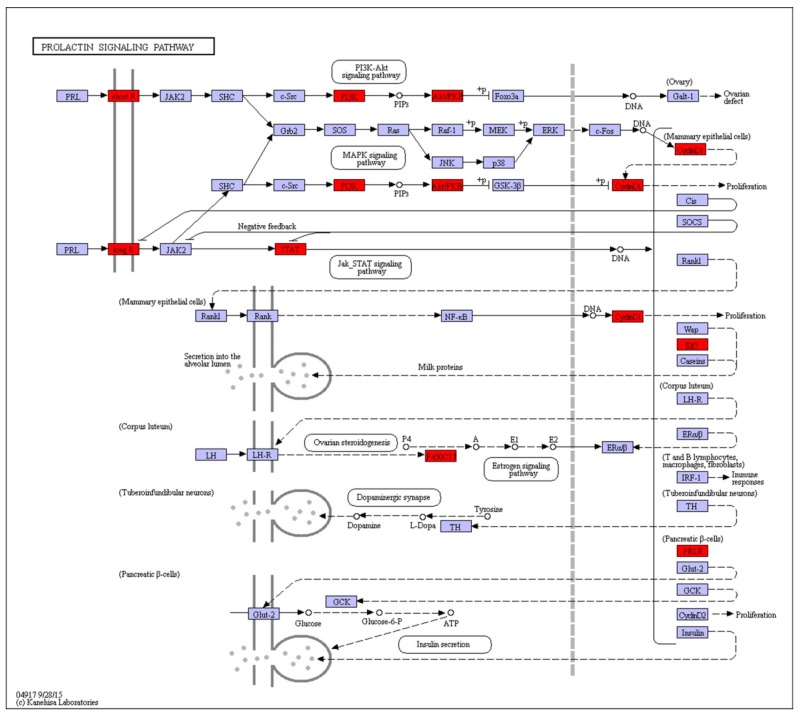
Model of activation of the prolactin signalling pathway in the *mink* uterus in activated versus diapause states. Differentially expressed genes are shown in red color. Genes in blue boxes were present in the *mink* transcriptome but not differentially expressed.

**Figure 4 ijms-20-02099-f004:**
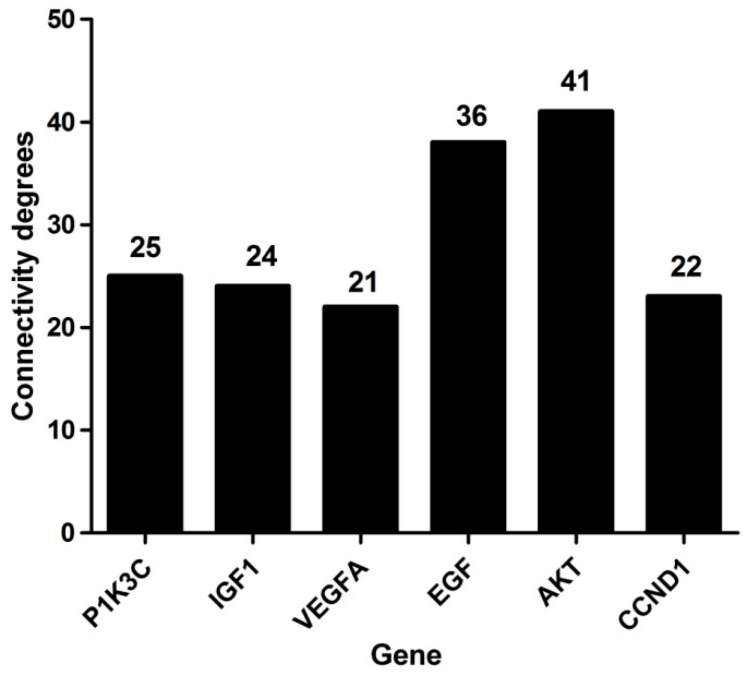
Connectivity degrees of the Protein-protein interactions (PPI) network.

**Figure 5 ijms-20-02099-f005:**
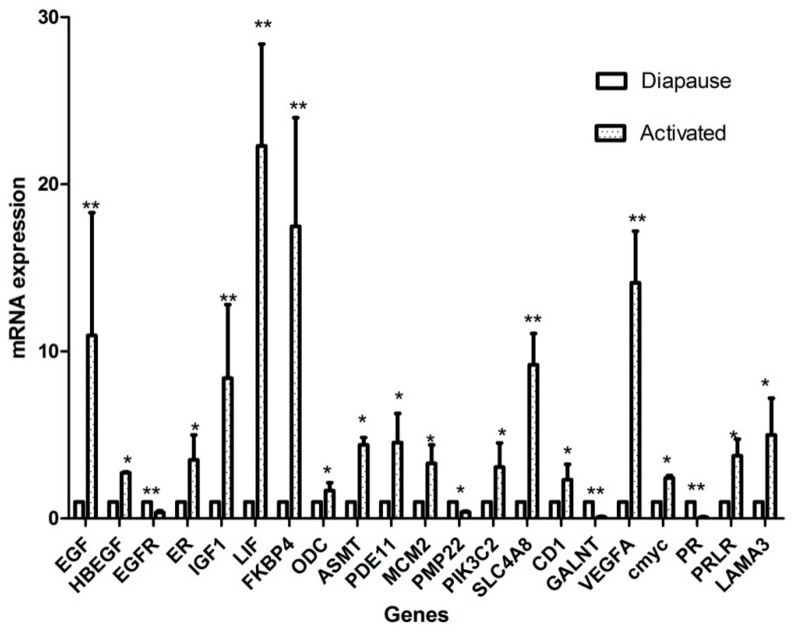
QRT-PCR validation of RNA-Seq data in *mink* uterus during embryo diapauses and activation: activated/diapause expression ratios. *GAPDH* was used as the internal control for mRNA analysis. The data shown are from three biological replicates. * *p* < 0.05, ** *p* < 0.01.

**Figure 6 ijms-20-02099-f006:**
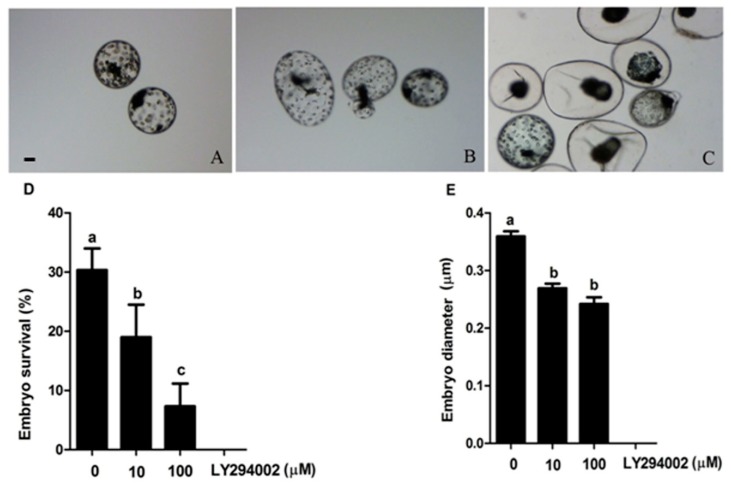
*Mink* embryos recovered in obligate diapause and cultured with different concentrations of the *PI3K/AKT* inhibitor (LY294002). The morphology of embryos was evaluated under a stereomicroscope. (×200) (**A**) Diapause embryos, (**B**) Expanded embryos indicative of reactivation, (**C**) Dead embryos, (**D**) Embryo survival after five days culture, (**E**) The mean diameter of the blastocysts after five days culture. Significant at *p* < 0.05. Data are mean ± SEM. Different small letters above columns indicate significant differences by Student’s *t*-test.

**Table 1 ijms-20-02099-t001:** Number of differential expressed genes (DEGs) in uterus of diapause and 5 days after embryo activation.

Comparison ^a^	Fold > 2 ^b^	Fold > 1.5 ^c^	*p* < 0.05 ^d^
Activated > Diapause	622 (738)	167 (178)	963 (1063)
Diapause > Activated	445 (502)	58 (67)	564 (621)
All	1067 (1240)	225 (245)	1527 (1684)

^a^ Gene expression changes. Activated > Diapause indicates that expression levels in activated are higher than during diapause. Diapause > Activated indicates that expression levels in activated are lower than diapause. ^b^ Genes with at least a two-fold change in expression between activated and diapauses. Number in parentheses indicates total number of DEGs. ^c^ Genes differential fold more than 1.5, and simultaneously differential fold less than 2 between the activated and diapause group. Number in parentheses indicates total number of DEGs. ^d^ Total number of differentially expressed annotated genes based on a *p*-value less than 0.05; number in parentheses indicates total number of DEGs.

**Table 2 ijms-20-02099-t002:** The top 20 most DEGs in the activated uterus compared with the diapause uterus (*p* < 0.05).

Gene Symbol	Log2 Fold Change	*p* Value	Gene Description
*ANKS1A*	7.9822	6.52 × 10^−^^42^	ankyrin repeat and SAM domain-containing protein 1A isoform X2
*TEAD*	−7.7671	3.41 × 10^−37^	Tead3 protein
*LAMA3*	7.4272	2.01 × 10^−33^	laminin, alpha 3
*XPO5*	−7.0783	8.76 × 10^−31^	XPO5 protein
*PNKD*	6.9436	1.04 × 10^−27^	probable hydrolase PNKD isoform X2
*N4BP1*	−6.5331	3.17 × 10^−34^	NEDD4-binding protein 1
*PDPR*	6.4222	6.67 × 10^−29^	pyruvate dehydrogenase phosphatase regulatory subunit
*CPSF3*	6.3617	4.02 × 10^−21^	cleavage and polyadenylation specific factor 3
*RIOK3*	6.3151	6.04 × 10^−23^	RIO kinase 3
*THOC1*	6.2389	3.67 × 10^−22^	THO complex 1
*RP11-489N6*	6.1613	4.70 × 10^−21^	BAC RP11-489N6
*LOC101694508*	−6.037	1.99 × 10^−19^	uncharacterized LOC101694508
*DSG2*	6.033	3.31 × 10^−28^	desmoglein 2
*SLC7A2*	6.0189	2.64 × 10^−17^	solute carrier family 7 (cationic amino acid transporter, y)
*ASPM, ASP*	5.9385	6.62 × 10^−18^	asp (abnormal spindle) homolog, microcephaly associated
*ATP6N*	5.9165	1.29 × 10^−18^	V-type proton ATPase
*ZXDC*	5.9001	3.30 × 10^−19^	ZXD family zinc finger C
*PDIA6*	5.6566	6.67 × 10^−17^	disulfide-isomerase A6
*PINK1*	−5.6508	2.77 × 10^−17^	serine/threonine-protein kinase PINK1, mitochondrial
*ERCC5*	−6.0343	6.47 × 10^−20^	immunoglobulin-like variable motif containing (BIVM)

**Table 3 ijms-20-02099-t003:** Top 10 gene ontology (GO) terms and enriched genes in the activated uterus vs. the diapause uterus at *p* < 0.01.

Category	Terms	*p* Value	Genes
BP	Xenobiotic metabolic process	2.98×10^−5^	*ITGB4↓, ACP2↓, LAMA3_5↑, ADAMTS9↑, PTPRF↑, COL12A1↑, TN↓, CHL1↑, NCAM↑, EPHB6↑, serB↑, LIFR↓, PRLR↑, PSPH↑, IGDCC4↓, CSF2RB↑*
	Cell communication	6.38×10^−5^	*RASL11B↑, SLC18A2↑, HDHD2↑, ANK↓, IKBIP↑, NEMF↓, DNMBP↓, ARHGAP12↑, LOXL2_3_4↓, ATP13A1↑, TN↓, IGF1↑, BBC3↓, STAC↑, PKN↑, ANPRA↓, SYTL↑, IL1A↑, IFT27↓, CCRL1↑, PSD4↑, RASA2, CD44↓, NOTCH4↑, ARL4A↓, RCAN1↑, ELAC2↓, DDIT4L↓, BIVM↓, PLCD1↑, ZDHHC21↑, CXCL14↓, KIAA1324↑, LAMA1↑……….*
	Signal transduction	7.08×10^−5^	*SPAG5↑, TNFSF18↑, ARHGAP6↓, LCP1↑, SIPA1L2↑, ITGB4↓, VIPR2↑, LGR5↓, NAV1↓, SYT4↑, THOC1↑, ZYX↑, DMBT↓, PIKFYVE↓, LRP2↑, RAB36↑, RASAL1↓, COL12A1↑, EPHB6↑, SPOCK↓, PPP2R3↑, STAT5A↓, NR4A1↑, CSF2RB↑, SIRPA↓…….*
	Cellular response to stimulus	9.02×10^−5^	*PDE7↑, NFKBIZ↓, PRSS15↓, KIAA1324↑, IGDCC4↓, PDZK1↑, TTC39A↑, RFX1_2_3↓, SEPT3_9_12↑, SLC41A↑, KIAA1324↑, CASR↓, GDF3↑, PLCB↑, NOD2↓, VEGFA↑, STAT2↑, PDE1↑, GPR143↑, ADCYAP1R1↓, THOC1↑, IGSF10↓, EGF↑, PDZD8↑…….*
CC	Extracellular matrix	7.05×10^−9^	*COL1AS↑, ATRNL1↑, COL4A↑, ADAMTS9↑, GPC6↑, LCP1↑, SPARC↑, PLS1↑, PPP2R3↑, SPOCK↓, SPTAN1↑, COL12A1↑, COL6A↓, ADAM33↓, PLS1↑, PLCD↑, SPARC↑, ADAMTS12↑, COL1AS↑,*
	Collagen trimer	7.79×10^−5^	*COL1AS↑, COL4A↑*
MF	Phosphatase activity	8.98×10^−6^	*PTPRF↑, ARL4↓, ALCAM↑, ACP2↓, EPHB6↑, CPZ↑, NCAM↑, COL12A1↑, PTP4A↑, SERB↑, CHL1↑, PHOSPHO2↑, EPHB6↑, DUSP↑, PTPRR↑, PRLR↑, CSF2RB↑, ADAMTS9↑, CCBL↑, ITGB4↓, DCHS1_2↑*
	N,N-dimethylaniline monooxygenase activity	4.28×10^−4^	*FMO↑, IL4I1↑*
	G-protein coupled receptor activity	4.42×10^−4^	*GPR143↑, GPR161↓, PACAPRI↓, TES↑, CNR1↑, VIPR2↑, MRGPRF↓, TMEM161B↑, NSUN2↑, CCRL1↑, PDZK1↑, FZD5_8↑, CASR↓, LGR5↓, KIAA1324↑, ADRA2C↓, RGS3↑, PAMR1↓, PDZD8↑, LGR5↓, HRH1↓*
	Metallocarboxypeptidase activity	7.08×10^−4^	*CPXM1↑, CPZ↑, CPN1↑, CPXM2↑*

**Table 4 ijms-20-02099-t004:** Most enriched pathways in the activated uterus vs. the diapause uterus at *p* < 0.01.

KEGG Pathway	*p* Value	Genes
ECM-receptor interaction	1.61×10^−11^	*COL1AS↑, CD44↓, LAMA1_2↑, TN↓, LAMA3_5↑, ITGB4↓, LAMC3↑, THBS1↓, COL6A↓, ITGA5↓, ITGB8↓, ITGA9↓, COL4A↑, CHAD↑, THBS2S↓*
Focal adhesion	7.53×10^−08^	*PIK3C↑, COL1AS↑, LAMA1_2↑, TN↓, LAMA3_5↑, FLNA↓, EGF↑, ITGB4↓, ROCK2↑, VEGFA↑, THBS1↓, PDPK1↑, ACTB_G1↑, COL6A↓, IGF1↑, ITGA5↓, AKT↑, ITGB8↓, LAMC3↑, PPP1C↑, ITGA9↓, COL4A↑, CHAD↑, THBS2S↓, CCND1↓*
PI3K-Akt signaling pathway	4.94×10^−7^	*PIK3C↑, COL1AS↑, LAMA1_2↑, COL3A1↑, TN↓, LAMA3_5↑, VEGFA↑, PPP2R3↑, MDM2↓, EGF↑, ITGB4↓, PRKAA↓, THBS1↓, PRLR↑, PKN↑, COL6A↓, IGF1↑, ITGA5↓, AKT↑, ITGB8↓, PDPK1↑, YWHAB_Q_Z↑, NR4A1↑, LAMC3↑, PHLPP↑, ITGA9↓, COL4A↑, CHAD↑, SGK1↑, THBS2S↓, FGFR4↑, CCND1↓*
Cytokine-cytokine receptor interaction	2.37×10^−2^	*VEGFA↑, TNFSF12↓, IL13RA1↓, CXCL14↓, CSF2RB↑, PRLR↑, IL1A↑, EGF↑, TNFSF18↑, LIFR↓, LTBR↓, BMPR1B↑*
Protein digestion and absorption	7.65×10^−6^	*COL1AS↑, ACE2↑, XPNPEP2↑, COL6A↓, SLC7A9↓, COL4A↑, SLC15A1↑, ACEH↑*
AGE-RAGE signaling pathway in diabetic complications	6.35×10^−5^	*PIK3C↑, COL1AS↑, PLCB↑, VEGFA↑, PLCD↑, IL1A↑, STAT5A↓, AKT↑, COL4A↑, CCND1↓*
Cell adhesion molecules (CAMs)	1.69×10^−3^	*MHC2↓, CLDN↑, ITGB8↓, ALCAM↑, CDH4↑, NCAM↑, PTPRF↑, SIGLEC1↓, ITGAM↓, ITGA9↓, PTPRC↑, NRCAM↑*
Glycine, serine and threonine metabolism	3.54×10^−3^	*GATM↑, serB↑, SARDH↑, CTH↑, gcvT↑, DAO↑, DLD↑*
Thyroid hormone synthesis	6.34×10^−3^	*PLCB↑, GPX8↑, TTF2↑, SLC26A4↑, LRP2↑, ADCY7↓, PAX8↑*
Platelet activation	0.92×10^−2^	*ROCK2↑, COL1AS↑, AKT↑, PIK3C↑, PLCB↑, PPP1C↑, ACTB_G1↑, ADCY7↓*
Prolactin signaling pathway	0.50×10^−2^	*ELF5↑, PIK3C↑, AKT↑, CYP17A↑, PRLR↑, STAT5A↓, CCND1↓*

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
