# Peer review of "Transcriptome Changes in the Mink Uterus during Blastocyst Dormancy and Reactivation"

_ijms, 2019, doi:10.3390/ijms20092099_

Round 1
Reviewer 1 Report
This paper describes the analysis of high throughput transcriptome data of uterine samples from mink on diapause and activation phase of the embryo development. The authors also cultured embryos and blocked the PI3K/ATK pathway inhibitor and found that the inhibition of PI3K/ATK pathway significantly reduces embryo survival. The paper is well written and organized in a very logical way that facilitates the reading. The figures and tables are well organized and represent the results well.
I have made some comments below that the authors need to address prior to having my recommendation for the publication of their work
Line 48: “average error rate of only 1-1.5%”. What does that mean? I do not understand what this adds to the introduction.
Line 49: “an unbiased method”. There are several reports showing that RNAseq presents results in a biased representation of transcripts. So I would suggest that this segment is removed.
Line 74: “clean reads”. I suggest that this is replaced by a more scientific term to state that the reads were trimmed for the removal of adapters and filtered for low quality reads.
Lines 79 80: “(622 genes fold > 2, and 167 genes fold >1.5)”, “(445 genes fold > 2, and 58 genes fold > 1.5)”. Table 1. This was confusing to me because most commonly we lower number of genes differentially expressed with higher thresholds.
On the legend of figures 1, 2 and 3, I suggest that you add “Model of” before the word activation.
On figures 1, 2 and 3, it was quite striking to me that there was not one gene with a non-colored rectangle. This is very unusual, especially for Kegg pathways, that are pre-established.
Figure 6 A-B, please add a scale bar
Subheading 4.1. You need to specify the number of samples here.
Line 374. “and paired-end reads” Add length please.
Line 376: “and low quality reads from” Indicate what criteria was used for the identification of reads.
Subheading 4.3. Please include the list of genes that was used as background for GO and KEGG testing. Also did you do any correction for multiple hypothesis testing?
Subheading 4.5. Which species did you use to identify orthologs, what database did you use to identify orthologs?
Author Response
Response to Reviewer 1 comments Thank you very much for your reply and help. We are grateful for the reviewers’ comments and kind suggestions for improvement of our manuscript. We provide this cover letter to explain, point by point, the details of our revisions in the manuscript and our responses to the reviewers’ comment. In the revised paper, we marked the revision with blue color. We hope you will now find the paper suitable for publication. Point 1: Line 48: “average error rate of only 1-1.5%”. What does that mean? I do not understand what this adds to the introduction. Response 1: Line 48: We have removed this sentence. (Now line 49) Point 2: Line 49: “an unbiased method”. There are several reports showing that RNAseq presents results in a biased representation of transcripts. So I would suggest that this segment is removed. Response 2: Line 49: We have removed this sentence. (Now line 49) Point 3: Line 74: “clean reads”. I suggest that this is replaced by a more scientific term to state that the reads were trimmed for the removal of adapters and filtered for low quality reads. Response 3: Line 74: Clean reads were obtained by removing reads containing adapter, reads containing ploy-N and low quality reads from raw data. We have read a large amount of literature on transcriptome sequencing studies, most of paper describe high quality reads as clean reads, also a few papers use the terms processed reads, nomorlised counts, high quality reads, filtered or trimmed reads. We tried to find a more scientific term to describe this high quality reads, but we are not sure which is the most suitable. If you have any better suggestions, please let me know. We are extremely appreciative of your advice. Point 4: Lines 79 80: “(622 genes fold > 2, and 167 genes fold >1.5)”, “(445 genes fold > 2, and 58 genes fold > 1.5)”. Table 1. This was confusing to me because most commonly we lower number of genes differentially expressed with higher thresholds. Response 4: Lines 79, 80: “d” indicates total number of differentially expressed annotated genes based on a P value of less than 0.05. Numbers in parenthesis indicate the total number of differentially expressed transcripts, including unannotated genes. “b” indicates number of differentially expressed annotated genes with 2-fold or greater difference between diapauses and activated uterus. Numbers in parenthesis indicate the total number of differential expressed transcripts, including unannotated genes. “c” indicates number of differentially expressed annotated genes with 1.5 fold or greater difference between diapause and activated uterus. Numbers in parenthesis indicate the total number of differential expressed transcripts, including unannotated genes. That is, d = b + c + genes with a differential fold less than 1.5. b = genes with a differential fold more than 2. C = genes with a differential fold more than 1.5, and simultaneously differential fold less than 2. For all these DEGs cut off is 0.05. Point 5: On the legend of figures 1, 2 and 3, I suggest that you add “Model of” before the word activation. Response 5: Lines 127, 131, 135, we have added “model of” before the word activation. (Now line 130, 134, 138) Point 6: On figures 1, 2 and 3, it was quite striking to me that there was not one gene with a non-colored rectangle. This is very unusual, especially for Kegg pathways, that are pre-established. Response 6: Figure 1, 2 and 3, We have revised the legends. Point 7: Figure 6 A-B, please add a scale bar Response 7: Figure 6: we have added a scale bar in the picture. Point 8: Subheading 4.1. You need to specify the number of samples here. Response 8: Subheading 4.1: we have added the number of samples in the subheading 4.2. (Now line 353-354) Point 9: Line 374. “and paired-end reads” Add length please. Response 9: Line 374: We have added the length of paired-end reads (Now line 364) Point 10: Line 376: “and low quality reads from” Indicate what criteria was used for the identification of reads. Response 10: Line 376: The number of bases that had a phred quality value ≦20 account for 50% of total reads, designed as low quality reads. Point 11: Subheading 4.3. Please include the list of genes that was used as background for GO and KEGG testing. Also did you do any correction for multiple hypothesis testing? Response 11: Subheading 4.3. We have listed top ten GO and KEGG terms and enriched differential expressed genes (Table 3 and 4). For GO and pathway enrichment analysis, P<0.05 was considered to indicate a statistically significant difference. Furthermore, P-values were adjusted using the false discovery rate (FDR) method for multiple hypothesis testing. FDR<0.05 was established as the threshold value. We have added the method in description of the statistical analyses (Now line 415-417). Point 12: Subheading 4.5. Which species did you use to identify orthologs, what database did you use to identify orthologs? Response 12: Subheading 4.5.The human protein reference database was used to analyze the PPI of DEGs, using String database to identify orthologs. We feel grateful for the reviewers’ excellent comments and your help on our manuscript. Best wishes, Sincerely yours, Xinyan Cao
Reviewer 2 Report
In this manuscript, the authors report the differently expressed genes in dormant and activated uteri in mink. They explored interesting information in uterine physiological clues. It is well wrote. However, followings should be considered.
1) You should give full information for the uterine sampling for example which day samples were used in RAN-seq.
2) Needed more information in prolactin and PI3K pathway....
3) Discussion looks very long. It is needed little bit simplify..
Author Response
Response to Reviewer 2 comments Thank you very much for your reply and help. We are grateful for the reviewers’ comments and kind suggestions for improvement of our manuscript. We provide this cover letter to explain, point by point, the details of our revisions in the manuscript and our responses to the reviewers’ comment. In the revised paper, we marked the revision with blue color. We hope you will now find the paper suitable for publication. Point 1: You should give full information for the uterine sampling for example which day samples were used in RAN-seq. Response 1: For transcriptome analysis, we collected the uterine samples during embryo diapause (diapause group) and on the sixth day after prolactin-induced embryo reactivation (activated group) (Now line 351-352) . Point 2: Needed more information in prolactin and PI3K pathway.... Response 2: We have added enriched genes of prolactin pathway in Table 4, and added some discussion in conclusion (Now line 430-433). Point 3: Discussion looks very long. It is needed little bit simplify Response 3: We have deleted some of the discussion. We feel grateful for the reviewers’ excellent comments and your help on our manuscript. Best wishes, Sincerely yours, Xinyan Cao
Round 2
Reviewer 1 Report
I am still stuck at this:
(622 genes fold > 2, and 167 genes fold > 1.5) and the same is valid for
(445 genes fold > 2, and 58 genes fold > 1.5)
My problem here was no that I misunderstood what they mean. I do not understand how it is possible that you have greater number of DEGs by applying stricter threshold.
Author Response
Response to Reviewer 1 comments
We feel great thanks for your professional review work on our manuscript.
Point 1: I am still stuck at this:
(622 genes fold > 2, and 167 genes fold > 1.5) and the same is valid for
(445 genes fold > 2, and 58 genes fold > 1.5)
My problem here was no that I misunderstood what they mean. I do not understand how it is possible that you have greater number of DEGs by applying stricter threshold.
Response 1: Fold > 2 indicates number of differentially expressed genes with 2-fold or greater difference between diapauses and activated uterus. Fold > 1.5 indicates a differential fold more than 1.5, and simultaneously differential fold less than 2. We have revised the result (Now line 81-82, and line 89-90).
We feel grateful for the reviewers’ excellent comments and your help on our manuscript.
Best wishes,
Sincerely yours,
Xinyan Cao
